# Barriers and facilitators associated with implementing interventions to support oral anticancer agent adherence in academic and community cancer center settings

**Benyam Muluneh**[1,2]*, **Michele A. Muir**[1], **James Bernard Collins**[1], **Darrian Proco**[1], **Emily Mackler**[3], **Ashley Leak Bryant**[2,4], **William A. Wood**[2,5], **Michael Tilkens**[6], **Jeffrey S. Reichard**[6], **Matthew Foster**[2,5], **Justin Gatwood**[7], **Stephanie B. Wheeler**[2,8], **Leah L. Zullig**[9,10], **Jennifer Elston Lafata**[2,11]

1 Division of Pharmacotherapy and Experimental Therapeutics, Eshelman School of Pharmacy, University of North Carolina at Chapel Hill, Chapel Hill, North Carolina, United States of America, 2 Lineberger Comprehensive Cancer Center, University of North Carolina at Chapel Hill, Chapel Hill, North Carolina, United States of America, 3 Michigan Oncology Quality Consortium, Ann Arbor, MI, United States of America, 4 School of Nursing, University of North Carolina at Chapel Hill, Chapel Hill, North Carolina, United States of America, 5 School of Medicine, University of North Carolina at Chapel Hill, Chapel Hill, North Carolina, United States of America, 6 Department of Pharmacy, University of North Carolina Medical Centerat Chapel Hill, Chapel Hill, North Carolina, United States of America, 7 College of Pharmacy, University of Tennessee Health Science Center at Memphis, Memphis, Tennessee, United States of America, 8 Department of Health Policy and Management, Gillings School of Global Public Health, University of North Carolina at Chapel Hill, Chapel Hill, North Carolina, United States of America, 9 Department of Population Health Sciences, Duke University School of Medicine, Durham, NC, United States of America, 10 Center of Innovation to Accelerate Discovery and Practice Transformation, Durham Veterans Affairs Health Care System, Durham, NC, United States of America, 11 Division of Pharmaceutical Outcomes and Policy, Eshelman School of Pharmacy, University of North Carolina at Chapel Hill, Chapel Hill, North Carolina, United States of America

* bmuluneh@unc.edu

**Data Availability Statement:** The data cannot be shared due to ethical restrictions by our institutional review board. Participants in this study

## Abstract

### Purpose

The goal of this study is to determine barriers and facilitators to the implementation of medication adherence interventions to support cancer patients taking novel, targeted oral anticancer agents (OAAs).

### Methods

We conducted qualitative interviews using a semi-structured guide from the Consolidated Framework for Implementation Research (CFIR). We used purposive sampling to identify clinicians (physicians, pharmacists, nurse practitioners, nurses) and administrators (leadership from medicine, pharmacy, and nursing) who delivered care and/or oversee care delivery for patients with chronic leukemia prescribed an OAA.

### Results

A total of 19 individuals participated in an interview (12 clinicians and 7 administrators), with 10 primarily employed by an academic cancer center; 5 employed by the community cancer

provided consent. During their consent, they were assured their identity would not be revealed. We also specified to the IRB that other than the quotes used in the publication, the entire transcript would be stored securely in a password protected computer to ensure the participant's identity would not be revealed. Requests for data access can be sent to the Institutional Review Board at University of North Carolina. Email is: irbis@unc.edu.

**Funding:** The project described was supported by the National Center for Advancing Translational Sciences, National Institutes of Health, through Grant KL2TR002490. The content is solely the responsibility of the authors and does not necessarily represent the official views of the NIH. GSK provided support in the form of salaries for authors MM DP, *and JC* but did not have any additional role in the study design, data collection and analysis, decision to publish, or preparation of the manuscript. The specific roles of these authors are articulated in the 'author contributions' section. There was no additional external funding received for this study.

**Competing interests:** I have read the journal's policy and the authors of this manuscript have the following competing interests: Benyam Muluneh is a consultant for Servier and his spouse is employed by Novartis; Michele Muir *and JB Collins are* a post-doctoral fellow with GlaxoSmithKline, Emily Mackler has research funding with Astra Zeneca; William Wood has research funding with Genentech and Pfizer; Leah Zullig has research funding with Proteus Digital Health, PhRMA Foundation and is a consultant for Novartis and Pfizer;

center; and 4 employed by the integrated health-system specialty pharmacy. Barriers identified included low awareness of adherence interventions, difficulty in adherence measurement, complexity of designing and implementing a structured adherence intervention, and competing priorities. Facilitators identified included support of hospital administrators, value for pharmacists, and willingness to embrace change. Participants also made recommendations moving forward including standardizing workflow, designating champions, iterating implementation strategies, and improving communication between clinicians and with patients.

## Conclusion

Individual and system level factors were identified as determinants of implementation effectiveness of medication adherence interventions. A multidisciplinary advisory panel will be assembled to design comprehensive and actionable strategies to refine and implement a structured intervention to improve medication adherence in cancer patients.

## Introduction

Oral anticancer agents (OAAs) have revolutionized the treatment of many cancer subtypes. One critical predictor of response to OAA treatment is near-perfect medication adherence (i.e, >90–95% of doses taken) [1–3]. Although adherence to these medications in clinical trials is >90%, adherence drops to 40% [2,4] in real-world settings due to barriers such as adverse drug reactions, cost, and forgetfulness. Several professional societies—including the American Society of Clinical Oncology (ASCO/NCODA), Hematology Oncology Pharmacy Association (HOPA), and Oncology Nursing Society (ONS)—have published consensus-based "best practice" standards to guide clinicians on optimal approaches of caring for patients on OAAs [5–7]. Attempts to adapt these standards within adherence interventions have been effective in improving patient adherence [8]. However, despite their clinical success, many OAA interventions have not been sustainable, highlighting how a lack of clear implementation strategies continues to serve as a barrier to OAA interventions [9–11].

To address this challenge, we developed implementation strategies for an OAA intervention using a systematic approach called Implementation Mapping [12]. In order to identify and develop these implementation strategies, we first needed to identify potential barriers and facilitators for an OAA intervention's success. Here, we report findings from the first phase of our study focused on determining existing barriers and facilitators to the implementation of adherence interventions in diverse clinical settings (academic and community cancer settings).

## Methods

### Study design

We conducted semi-structured interviews to identify implementation barriers and facilitators of a structured OAA adherence intervention. We defined a structured OAA adherence intervention as a program that consists of proactive education and monitoring of patients at pre-specified time points. The research team designed and pilot tested an interview guide developed by adapting questions available from the Consolidated Framework for Implementation Research (CFIR)'s online toolkit [13].

## Study setting and participants

We conducted this research within a large health system that comprises both academic and community care settings within the United States. This health-system has a medically integrated electronic health record, which allows all patient documentation, care team communication, and medication workflows to be conducted within it. Characteristics of patients treated at both hospitals include a median annual household income of $54,602, 31.3% of people have achieved a bachelor's degree or higher, 22.2% of people are uninsured, and 14.7% of people are living below the poverty line. Interviews were conducted at the academic cancer center and a community hospital. Additionally, we interviewed pharmacy staff at an integrated health-system specialty pharmacy that services all hospitals and clinics within the health system. We chose these two sites due to differences in resources addressing OAA adherence. Interviewees from the academic cancer center and health-system specialty pharmacy included individuals involved in a previous OAA pilot while interviewees from the community hospital were not. We used purposive sampling to identify clinicians (physicians, pharmacists, nurse practitioners, nurses) and administrators (leadership from medicine, pharmacy, and nursing) known to the research team who delivered care and/or oversee care delivery for patients on an OAA. Snowball sampling [14] was used to identify additional stakeholders.

## Data collection

Data were collected from August 2021 to October 2021. We obtained verbal consent from all participants before commencing the interview. All interviews were completed one respondent at a time via video teleconference. Using the semi-structured interview guide, after briefly describing the structured OAA intervention, we elicited respondents' perceptions of barriers and facilitators to implementing the structured adherence intervention in their practice environment. The interview guide focused on the following CFIR constructs [13]: characteristics of the adherence monitoring program, outer setting, inner setting, individual factors, and process factors (Appendix B). Interviews were conducted by one member of the research team, with a second member taking notes. The content of the interview guides was written and conveyed in English and was designed to provide sufficient flexibility, ensuring coverage of all necessary research domains while allowing investigators to explore novel topics that emerged during the interview. All interviews were audio-recorded using the video teleconferencing software Zoom (San Jose, CA: Zoom Video Communications Inc). Prior to coding, we used Zoom's automatic transcription service to transcribe the interviews verbatim. All transcripts were edited for accuracy by one investigator.

## Data analysis

We coded data using thematic content analysis relying on a priori defined CFIR constructs [13]. We synthesized findings by intrapersonal, interpersonal, and organizational level barriers and facilitators to identify the level of influence for each barrier and facilitator. We used ATLAS.ti software (Atlas.ti 8 Windows) to organize data for thematic coding and analysis. To perform quality assurance, two investigators double-coded one-fifth of the interviews at periodic intervals throughout the coding process. Inter-rater reliability was calculated, and non-concordance between coders was resolved by a third investigator. Methods are reported per the consolidated criteria for reporting qualitative research framework [15]. The design and procedures for research were reviewed and approved by the investigational review board at the University of North Carolina at Chapel Hill.

# Results

## Participant characteristics

A total of 19 individuals participated in an interview (12 clinicians (CLs) and 7 administrators (ADs)), with 10 primarily employed by the academic cancer center (3 pharmacy managers, 2 cancer center directors, 2 oncologists, 1 outpatient clinical pharmacist, 1 nurse practitioner, and 1 nurse navigator); 5 employed by the community cancer center (2 oncologists, 2 patient education nurses, 1 inpatient clinical pharmacist); and 4 employed by the integrated health-system specialty pharmacy (2 pharmacy managers and 2 specialty pharmacists). Barriers and facilitators (along with illustrative quotes), as well as proposed solutions by interviewees are outlined below and briefly summarized in Tables 1 and 2. Barriers and facilitators have been designated as "current" or "anticipated" and are presented according to the CFIR classification system.

## Barriers

**1. Intervention characteristics.** *1.1 Complexity*: *Difficulty in measuring adherence*. Clinicians expressed difficulty in being able to measure adherence accurately and efficiently, reporting they primarily assess adherence via patient self-report, typically without using a standardized questionnaire. Additionally, other means of quantifying adherence such as in-clinic pill counts were also reported as being suboptimal. *"We really have no way of actually monitoring adherence, because we do not ask the patients to bring in their medications so I can count them. Ideally that would be great, but when we did it in real life. . .I was petrified I was going to drop the pills on the counter." CL10*

Furthermore, participants expressed low awareness of the most optimal approach for obtaining accurate adherence information from patients. As illustrated by the following comment, lack of awareness regarding an optimal adherence assessment approach may be contributing to patients being less forthcoming about their missed doses. *"But I personally probably feel like patients are not disclosing all of the times that they actually miss doses. And I don't know if it's because of the way we're asking the question."AD1*

Additionally, participants mentioned that pharmacy technicians—who call patients monthly to arrange refills—often ask about missed doses but are not adequately trained address nonadherence. *"And I know right now we don't really have, like an algorithm for technicians to, you know, collect that data. And maybe the patient says they missed two doses of their medicine. What does a technician do with that?" AD1*

*1.2 Complexity*: *Poor communication with external specialty pharmacies*. Additionally, interview participants raised significant challenges in OAA adherence monitoring due to poor communication between external specialty pharmacies and the patient care team. Participants lamented that automated adherence information from external specialty pharmacies is not feasible because these pharmacies are not integrated into the health system's electronic medical records nor have optimized workflows to designed to the patient care team (e.g., Epic Systems Corporation). *"Somehow, you'd have to link the outside specialty pharmacies with Epic because there's no way to make it automated if it's a completely different system. I don't know if that's possible, but I mean, if I'm getting a fax from these people, I'm not seeing it." CL3*

**2. Outer setting: Patient needs and resources.** There are also several patient-level and socioeconomic barriers to OAA access that contributes to nonadherence. Participants noted factors such as affordability, health literacy, and cognitive impairment being insufficiently addressed by current processes. For example, one participant noted that although there are medication assistance programs, these may not be reliable at all times of the year. *"I think other things that we've seen are just the ability to afford a medication. We've certainly got programs in*

**Table 1. CFIR domains and constructs.**

| CFIR Domains | Constructs [13] | | | Illustrative Quotes |
|---|---|---|---|---|
| Intervention Characteristics | Adaptability (Facilitator) | | | "You want this to work in a busy community practice community practicing setting." CL1 |
| | Complexity (Barrier) | | | "So what is coming into my clinic is becoming more and more complex. These are older adults on chemotherapy, on oral chemotherapy, on a combination of oral and infusion chemotherapy that we didn't have several years ago." CL9 |
| | Evidence Strength and Quality | | | "...there would be there would need to be a compilation of research data that supports that. In terms of improved adherence practices, improve disease states related to adherence." CL10 |
| | Trialability | | | "I think in the beginning...probably be informal and potentially if there was value to it, then it might be something more formal." CL2 |
| Outer Setting | Patient Needs and Resources (Barrier) | | | "I think other things that we've seen are just the ability to afford a medication. We've certainly got programs in place to help, but there are situations where a patient's co-pays, especially at the end of the year, are just prohibitive to them obtaining that medication." AD3 |
| Inner Setting | Implementation Climate | Compatibility | | "Just figuring out how to incorporate it into patient appointments...if you you're seeing 20 patients a day, are you the one who...helps to assess adherence or is there someone else who comes when you leave the patient's room?" CL10 |
| | | Relative Priority | | "But I know that our acute leuks take up so much of our time that the our chronics often fall to the wayside because they just we can put them a little bit more on autopilot." CL3 |
| | | Tension for Change | | "What we don't want to do is rush it and not making good decisions and then have it kind of fail because we didn't we didn't think of all of the things or plan and clean it out well." |
| | Networks and Communications | | | "...if [the patient] missed more than two doses in a month and I will contact the provider and let them know." CL12 |
| | Structural Characteristics | | | "So whether it's virtual or in person, it's another phone call or another visit that they come in to come in for. Some patients may appreciate that. Some may be resistant to it. Some providers may appreciate that, some providers may be resistant to it." AD5 |
| | Readiness for Implementation | Leadership Engagement | | "I think my colleagues would. I don't know for the administrators in terms of, you know, with limited resources." CL8 |
| | | Available Resources | | "I can't imagine there wouldn't be a lot of enthusiasm for this, to be honest." CL4 |
| | | Access to Knowledge and Information | | "And so if we want to do a very effective motivational interview with the patient to understand truly what their adherence barriers are, then we need more time to do that. And that would require us to have more resources.." CL3 |
| Characteristics of Individuals | Knowledge/Belief | | | "I'm not like perusing literature specifically for adherence data these days just because I'm not in a position yet to structure that program." CL9 |
| | Self-Efficacy | | | "I don't know if it's because of the way we're asking the question or if it's because they don't want to hear our response." AD1 |
| | Other Personal Attributes | | | "We as providers have a lot of the same discussions with patients, but our pharmacy team do it in a more regimented, formal manner." CL4 |
| Process | Champions | | | "I think we are still in a culture of you need a physician champion." |
| | Engaging | | | "...really working through our front line team, because I know that they all have really good thoughts about it and opinions and they're the ones actually doing the work." AD1 |
| | Executing | | | "My my ideal scenario is we have a clear list, whether it's a functionality, in EPIC or it's an outside of EPIC thing, but it's a clear formatted process as far as like how do I identify all my patients for an oral chemo and keep track of when my last touch points were." CL9 |
| | Planning | | | "I think SOPs would be needed to sort of standard operating policies and procedures." CL7 |
| | Reflecting/Evaluating | | | "We did an evaluation analysis...And this was one of the projects that came out of that." AD1 |

place to help, but there are situations where a patient's co-pays, especially at the end of the year, are just prohibitive to them obtaining that medication." AD3

Another participant noted how certain patients with low literacy skills may be unable to follow medication administration instructions on prescription bottle labels. "And the other thing is illiteracy... they get these pill bottles from the pharmacy that say take one pill once a day, this pill every other day. And no wonder that they have trouble complying because they have 10 pill bottles they don't read so well."AD4

**Table 2. Identified barriers and potential solutions.**

| Key Barriers | Potential Solutions |
|---|---|
| Complexity: difficulty in measuring adherence | Combined self-report and proportion of days covered (PDC) |
| | Educating clinicians on motivational interviewing |
| Complexity: poor communication with external specialty pharmacies | Designate personnel who proactively outreach to external pharmacies |
| Patient needs and resources | Streamline services offered by social work and medication assistant technicians |
| Structural characteristics: lack of standardization in encounters and documentation | Develop standardized scripting in the electronic medical record to prompt clinicians to gather and document key pieces of information |
| Implementation climate: relative priority | Integrate the intervention into current workflow recognizing existing competing priorities |
| Readiness for implementation: available resources | Justify resources needed to administrators |
| | Demonstrate return on investment to administrators |
| Knowledge/Belief: low awareness on designing structured adherence interventions | Provide intervention design and implementation science support to adopters and implementers |

**3. Inner setting.** 3.1 Structural characteristics: Lack of standardization in encounters and documentation. Participants noted challenges with coordinating patient encounters for adherence assessments with their existing provider visits. For example, one participant expressed concerns regarding acceptability by both patients and providers; *"So, whether it's virtual or in person, it's another phone call or another visit that they come in for. Some patients may appreciate that. Some may be resistant to it. Some providers may appreciate that, some providers may be resistant to it."* AD5

Participants also noted the need for infrastructure changes and/or documentation practices within the electronic health record to better support and automate the process by which patients on OAAs can be identified and tracked over time.

"My ideal scenario is we have a clear list, whether it's a functionality in Epic or it's an outside of Epic thing, but it's a clear formatted process as far as like how do I identify all my patients for an oral chemo and keep track of when my last touch points were?" CL9

3.2 Implementation climate. *3.2.1 Relative Priority.* Many of the increasing demands on healthcare providers may contribute to stress, and adherence monitoring may be easily deprioritized. As one participant expressed: *"I think all health care providers right now are so overwhelmed and adherence is really just one more thing that would be very easy to take for granted."* CL5

3.3 Readiness for implementation: Available resources. There are also challenges of making a financial case for focusing on medication adherence. Respondents noted how budgetary constrictions were worsened by the COVID-19 pandemic: "*With COVID, there are so many things that we wanted to do that have just been put on the back burner. . .the organization did not want to hire any resources.*"AD3. Investment of new resources would be challenging for a clinical service without clear financial justification. One administrator mentioned: "*You know, if it's not directly revenue generating, it's got to be providing value somewhere else. right? Back to our good old value equation if it's not revenue generating or cost savings.*" AD6

**4. Characteristics of individuals: Knowledge/Belief: Low awareness on designing structured adherence interventions.** Although there was general awareness of how adherence impacts patient outcomes, not all participants were aware of the magnitude of importance.

One participant noted: "*It's been a while since . . . I've thought about this. . .I haven't read up on any of the newer literature.*" CL1

Even though they may not have a specific method in mind, most participants expressed the need for monitoring and addressing adherence stating: *"Everyone agrees that adherence is an issue"* CL11.

## Facilitators

**1. Intervention characteristics.**  *1.1 Trialability*. Participants noted an anticipated facilitator would be a successful pilot. Generating data that show how the intervention and strategies are effective would garner support from a broader group of clinicians and administrators and facilitate adoption.

> "I think you'd have to pilot it somewhere and show improvement, you'd have to have something, something data-driven to tell you that it works, and it has a direct impact on patient clinical outcomes." AD6

*1.2 Adaptability*. Participants noted a critical need to consider adaptations of a structured adherence intervention depending on setting of implementation (i.e., academic medical center versus community practice). These adaptations are necessary due to the volume of patients and availability of resources in different settings. Determining core and adaptable components of the intervention can facilitate equitable delivery.

> *"You want this to work in a busy community practice community practicing setting."* CL1

**2. Inner setting.**  2.1 Structural characteristics. Participants noted that a potential facilitator would be to standardize and define workflow around care of patients on OAAs in the context of a structured adherence intervention. Roles and responsibilities would need to be prespecified—better defining traditional roles (i.e., physician, pharmacist, nurse) and considering how to involve medical assistants, trainees, and volunteers into the workflow for additional support.

> "Better optimize some of the existing roles, whether it be technician roles. . . residents, students getting them more involved in a structured format is definitely another opportunity." AD2

> "I think there's a lot of inefficiencies in the work that's done. And I think there's a lot of things, let's say, in the nurse world that could be potentially done by the [medical assistant] to elevate that nurse to be able to do more." AD7

2.2 Implementation climate: Support for pharmacists. All participants consistently supported pharmacists playing a key role in the implementation of a structured adherence program. Even in the community cancer center lacking clinical pharmacy support, clinicians expressed desire to further include on-site clinical pharmacists in their medical teams.

> "I think having pharmacists embedded in the clinics is a huge benefit where they can work side by side with the patients but with the providers and the nurses in terms of coordinating their care." AD2

2.3 Readiness for implementation. *2.3.1 Available resources*: *Internal specialty pharmacy with access to patient health records*. Once patients needing a follow-up encounter are

identified, electronic medical and pharmacy records are reviewed prior to the patient encounter. As demonstrated by the following quote, the health-system in-house specialty pharmacists have access to the cancer center's electronic health record, giving them an advantage compared with their external specialty pharmacy counterparts. There are a multitude of other benefits, including but not limited to a shared medication list, real-time communication with providers in the EHR, and standardized documentation to ensure patients are engaged around adherence issues by disease and medication.

"Any time we're doing an outreach call, we've asked our team to really review the patient's chart. . .Hopefully there's updated notes to just understand kind of what the patient's treatment plan is. I think we've seen that patients' treatments can change pretty quickly." AD3

2.4 Tension for change: Institutional willingness to embrace change. Even though implementation of a structured adherence intervention may require some workflow changes, clinicians and administrators expressed willingness to modify current practices for a better one that more effectively addresses medication adherence in cancer patients. Participants noted a uniquely "innovative culture" of the institution that serves as a positive implementation climate for a structured adherence intervention. One respondent stated: *"I think we're fortunate in our organization where I think most of the providers understand the benefits and I don't think would have much pushback. But other organizations that may not be the case."* AD2

**3. Characteristics of individuals.** 3.1 Knowledge/Belief (Participants broadly understand the importance of adherence). All clinicians expressed the critical importance of assessing and addressing adherence in patients on OAAs. Clinicians understand how non-adherence impacts short term and long-term patient outcomes. One participant noted: *"When patients are noncompliant or do not take their oral chemotherapy agents, this can lead to their cancer not responding, relapses, poor outcomes in their cancer care, and can also impact quality of life issues, obviously, if their cancer is coming back and or leading to complications."* CL4

Administrators also acknowledged the adverse implications of OAA non-adherence including data from the health-system suggesting the correlation of adherence on patient outcomes. *"I think [nonadherence] is a significant issue that probably leads to poor outcomes in patients who aren't adhering to the oral chemotherapy. I know we have internal data that shows some of those numbers. And I think there's good literature support that suggests that patients have poorer outcomes based on poor adherence."* AD2

**4. Process: Engaging: Champions.** Interview participants emphasized the importance of designating a champion to facilitate the implementation and sustainability of the adherence intervention. An ideal champion was described as being passionate about promoting adherence and, ideally, engaged in direct patient care. Participants did note that championing an adherence intervention to peer clinicians would be relatively straightforward given the general awareness about the importance of this topic.

> *"Ideally, if you had an adherence champion, it would be someone who's well versed in the in the medicines and. And the toxicities of it, but is also passionate about caring for patients. . .it'd be good to have an MD on board."* CL1

## Discussion

To design stakeholder-informed implementation strategies, we conducted qualitative interviews that yielded both current and anticipated barriers and facilitators that will need to be addressed, including factors at both individual (e.g., self-efficacy, knowledge, belief) and system (e.g., competing priorities, leadership engagement, resources) levels. Participants also offered several concrete recommendations that both mitigate barriers and leverage facilitators, including

improving institutional policies and procedures, standardizing adherence measures, and defining and tracking key intervention performance indicators to evaluate long-term success.

## Policies, procedures, and standardization

Participants noted the importance of having clear institutional policies and procedures outlining the operations behind a structured adherence intervention. Since many members of the healthcare team interact with the patient, it is important to specify roles and responsibilities. In addition to clinicians—pharmacists, nurses, and physicians—trainees, pharmacy technicians, and medical assistants may also play a supportive role in the maintenance of a structured adherence intervention. There were several aspects of the existing OAA workflow noted by interview participants. **Fig 1** depicts this workflow graphically to illustrate the processes and moderators that need to be considered for successful execution of the structured adherence intervention as recommended by participants.

Our participants noted the benefits of patients filling their OAAs at an integrated health-system pharmacy. Integrated health-system pharmacy services have been shown to improve adherence [16], improve patient satisfaction [17], and may even lead to improved patient outcomes [8]. The ASCO/NCODA and HOPA national standards have separately indicated the clinical, humanistic, and financial benefits of cancer centers offering specialty pharmacy services [6,7]. Insurance restrictions (e.g. CVS Caremark plans requiring their beneficiaries to fill at CVS Caremark pharmacy) create a major obstacle to a significant number of patients from filling their OAA at an internal specialty pharmacy. Structured adherence interventions would clearly need to adapt existing processes for patients who do not have access to internal specialty pharmacy services. Using a systematic approach such as ASCO's Quality Oncology Practice Initiative (QOPI) framework has also been shown to optimize the safe use of OAAs [18–20].

Participants also noted the critical need to keep social determinants of health (SDOH) in mind when initiating patients on OAAs. Socioeconomic barriers such as cost, lack of insurance, low health literacy, and experiencing homelessness were all noted as potential causes of suboptimal outcomes in certain vulnerable patients. Others have also found that SDOH creates

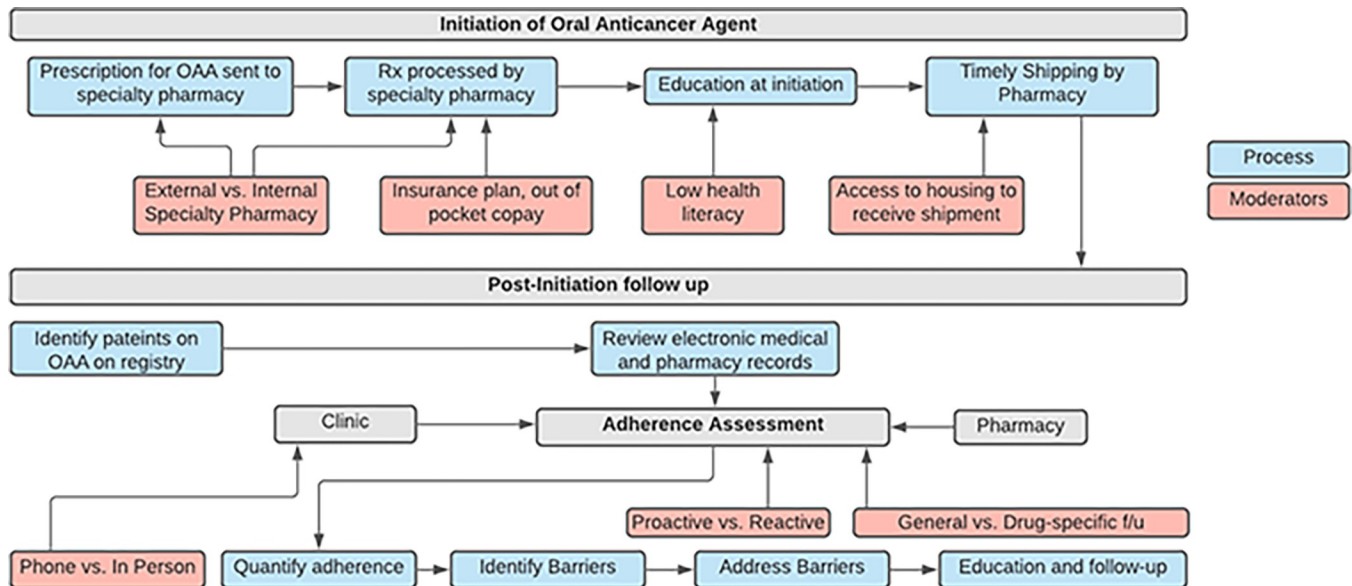

**Fig 1. Execution of a structured adherence program.** OAA = Oral Anticancer Agent. The initiation of OAAs as well as the post-initiation follow up process is displayed in blue. Important moderators that impact the quality, fidelity, timeliness, and completeness of each step in the process is displayed in red.

significant racial and ethnic disparities in medication adherence especially in the adjuvant breast cancer setting [21–23].

## Standardizing adherence assessment and training staff

Measuring adherence in the clinical context was identified as a key barrier. Most self-report adherence measures were designed for research purposes, minimizing their clinical utility. As such, there is a need to design and validate ways to monitor medication adherence in the clinical setting. There have been adherence measures that have been validated in other diseases including primary care and antiretrovirals in HIV/AIDS treatment that could be adapted [24]. Shorter adherence measures that take into account social desirability bias and ones that could be integrated into the EHR are most optimal [24,25]. During our initial pilot of a medication adherence program, we tested a one-item question ("since the last time we saw you, how many doses do you think you may have missed?"), which we validated with proportion of days covered (PDC) data [8]. However, PDC itself has some limitations and may overestimate adherence in cases where pharmacies are automatically sending medications to patients (thus falsely appearing as though the patient is taking the drug). In the clinical setting, a combination approach (e.g., self-report and PDC) may be useful [24].

Participants acknowledged the need for additional training to conceptualize and conduct adherence assessments with patients. Most clinicians lack extensive training in motivational interview (MI) techniques as part of their professional curricula. An important strategy may be educating clinicians (including technicians who often interact with patients) how to use MI to facilitate patient disclosure of non-adherence by building trust and minimizing judgement. Additionally, the EMERGE guidelines put forth by the International Society of Medication Adherence [26] can be leveraged to conceptualize adherence in phases: initiation (i.e. starting treatment at the intended time), implementation (i.e. proportion of doses taken as prescribed), and persistence (i.e. period of time spent on drug compared to intended duration).

## Registry/Dashboard for patients on OAAs

Once patients have initiated OAA treatment, participants noted the need for a registry to identify patients who need follow up monitoring. This was emphasized by both pharmacy and clinic personnel because there is currently not an optimal system to automatically keep track of patients on OAAs and schedule follow up encounters. Dashboards have been demonstrated to visualize the effectiveness and gaps of cancer care delivery interventions [27,28]. Additionally, there is a need to develop a risk stratification system (i.e., "high" or "low" risk for adherence), which could guide the development of strategies that are individualized.

## Design equitable implementation strategies

Additionally, continually assessing the intervention for equitable adaptation was discussed. Given the diverse needs of patients treated at the academic and community institutions, the intervention and implementation strategies would need to be agile enough to meet the needs of all patients. For example, virtual visits could be ideal for patients who may live far from the cancer center; however, there would need to be efforts to ensure patients who have poor broadband access are able to be reached by the intervention.

## Limitations

Our interviews were conducted at three unique sites (academic cancer center, community cancer center, and an integrated health-system integrated specialty pharmacy) but were all within

one large state health system that share the same electronic medical record. Generalizing findings to other unique settings such as private practice contexts may be difficult. Additionally, the community hospital was still located in a relatively urban area and may not be reflective of other community cancer centers located in rural settings and their associated populations. Despite these limitations, the academic medical center is the primary referral hospital for cancer patients in the state and does have many vulnerable patients who face numerous structural and systemic barriers. There may have been some sampling bias given the principal investigator's knowledge of the some of participants prior to the study; however, we mitigated this by conducting the interviews with research staff who had no prior connection to the study participants. Additionally, our purposive sampling approach allowed us to have optimal representation of key stakeholders to lend their voices to this study.

## Conclusion

Interviewed stakeholders identified key individual and systematic barriers and facilitators to implementation of a structured medication adherence intervention. A multi-disciplinary advisory panel will be assembled to design strategies to mitigate these barriers including: (1) developing standard operating procedures with defined roles and responsibilities; (2) training clinicians and staff on assessing and addressing adherence; and (3) process mapping clinician workflows to provide efficiency and clarity.

## Author Contributions

**Conceptualization:** Benyam Muluneh, Darrian Proco, Emily Mackler, Ashley Leak Bryant, William A. Wood, Michael Tilkens, Matthew Foster, Justin Gatwood, Stephanie B. Wheeler, Leah L. Zullig, Jennifer Elston Lafata.

**Data curation:** Benyam Muluneh, Michele A. Muir.

**Formal analysis:** Benyam Muluneh, Jennifer Elston Lafata.

**Funding acquisition:** Benyam Muluneh, Stephanie B. Wheeler, Leah L. Zullig, Jennifer Elston Lafata.

**Investigation:** Benyam Muluneh.

**Methodology:** Benyam Muluneh, Stephanie B. Wheeler, Leah L. Zullig, Jennifer Elston Lafata.

**Project administration:** Benyam Muluneh, Michele A. Muir, James Bernard Collins, Darrian Proco, Ashley Leak Bryant, William A. Wood, Michael Tilkens, Matthew Foster, Stephanie B. Wheeler, Jennifer Elston Lafata.

**Resources:** Benyam Muluneh, Jennifer Elston Lafata.

**Supervision:** Benyam Muluneh, Emily Mackler, Ashley Leak Bryant, William A. Wood, Michael Tilkens, Stephanie B. Wheeler, Leah L. Zullig, Jennifer Elston Lafata.

**Validation:** Benyam Muluneh, Jennifer Elston Lafata.

**Visualization:** Benyam Muluneh.

**Writing – original draft:** Benyam Muluneh.

**Writing – review & editing:** Benyam Muluneh, Michele A. Muir, James Bernard Collins, Emily Mackler, Ashley Leak Bryant, William A. Wood, Michael Tilkens, Jeffrey S. Reichard, Justin Gatwood, Stephanie B. Wheeler, Leah L. Zullig, Jennifer Elston Lafata.

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
