## [Decision Letter · Decision Letter 0]

6 Jan 2023

PONE-D-22-22462Barriers and facilitators associated with implementing interventions to support oral anticancer agent adherence in academic and community cancer center settingsPLOS ONE

Dear Dr. Muluneh,

Thank you for submitting your manuscript to PLOS ONE. After careful consideration, we feel that it has merit but does not fully meet PLOS ONE’s publication criteria as it currently stands. Therefore, we invite you to submit a revised version of the manuscript that addresses the points raised during the review process.

Please note that we have only been able to secure a single reviewer to assess your manuscript. We are issuing a decision on your manuscript at this point to prevent further delays in the evaluation of your manuscript. Please be aware that the editor who handles your revised manuscript might find it necessary to invite additional reviewers to assess this work once the revised manuscript is submitted. However, we will aim to proceed on the basis of this single review if possible.  Could you please revise the manuscript to carefully address the concerns raised by the reviewer?

We look forward to receiving your revised manuscript.

Kind regards,

Steve Zimmerman, PhD

Associate Editor, PLOS ONE

Journal Requirements:

“The project described was supported by the National Center for Advancing Translational Sciences, National Institutes of Health, through Grant KL2TR002490. The content is solely the responsibility of the authors and does not necessarily represent the official views of the NIH”

“I have read the journal's policy and the authors of this manuscript have the following competing interests: Benyam Muluneh is a consultant for Servier and his spouse is employed by Novartis; Michele Muir is a post-doctoral fellow with GlaxoSmithKline, Emily Mackler has research funding with Astra Zeneca; William Wood has research funding with Genentech and Pfizer; Leah Zullig has research funding with Proteus Digital Health, PhRMA Foundation and is a consultant for Novartis and Pfizer;”

We note that one or more of the authors are employed by a commercial company

5. Please include a caption for figure 1.

6. Please include your tables as part of your main manuscript and remove the individual files. Please note that supplementary tables (should remain/ be uploaded) as separate "supporting information" files.

Reviewers' comments:

Reviewer's Responses to Questions

**Comments to the Author**

1. Is the manuscript technically sound, and do the data support the conclusions?

Reviewer #1: Yes

2. Has the statistical analysis been performed appropriately and rigorously? 

Reviewer #1: N/A

3. Have the authors made all data underlying the findings in their manuscript fully available?

Reviewer #1: Yes

4. Is the manuscript presented in an intelligible fashion and written in standard English?

Reviewer #1: Yes

5. Review Comments to the Author

Reviewer #1: Dear Editors

Thank you for the opportunity to review the manuscript, “Barriers and facilitators associated with implementing interventions to support oral anticancer agent adherence in academic and community cancer center settings.” The manuscript is well written and utilizes an important implementation science framework – the Consolidated Framework for Implementation Research (CFIR) – to identify barriers and facilitators to addressing medication adherence. The application of CFIR allows the authors to group the factors into useful domains that have been shown to be associated with implementation effectiveness. However, my enthusiasm for the manuscript is dampened by the following: there is confusion around comments that refer to the hypothetical intervention and those that refer to the current status quo. I think this can be addressed by dividing the results into current barriers and facilitators to adherence and factors related to implementation of a hypothetical intervention. I also encourage reframing the approach to be more in line with factors thought to be important for implementation effectiveness of an adherence intervention. This may be considered a subtle difference but makes it more inline with the use of the CFIR to design interventions. The CFIR Is an extensive framework and the complete guide quite long. Were all questions and domains included or were some selected and if so that process needs to be included. The other point to make is that a description of the setting is important – which country, average income, access to healthcare services, insurance coverage, etc? I assume this is a US-based study but that needs to be made clear and the context better described.

ABSTRACT:

Methods: remove “known to the research team” doesn’t seem relevant, the acronym “CL” is not defined

Results: see comment above in dividing results to what are perceived current barriers/facilitators and factors associated with implementation effectiveness of an intervention – thinking more of using this to design an intervention. Consider discussing results by domain of CFIR.

Conclusion: “successful implementation” is different from implementation effectiveness. Be careful with use of terminology when using implementation frameworks.

INTRODUCTION

The acronym CL is not defined.

Last paragraph: I think framing this as current barriers/facilitators to adherence and adherence monitoring, plus determining factors that would contribute to implementation effectiveness.

Using a hypothetical intervention is difficult to follow unless the intervention is spelled out in detail (see methods section) and if the person is can fully visualize what that would look like day-to-day. Instead, more useful to frame this as information gathered on what people think would be helpful to see in an intervention to target adherence.

METHODS

The reference is made to prior interventional work on oral cancer therapy management. The introduction section expands on this intervention saying it was effective to some degree but not sustainable. Why were individuals that were part of that intervention not interviewed as opposed to interviewing individuals with a hypothetical intervention they were not involved with? This paper would have been more informative using implementers from the pilot project. If that was not possible that should be explained to provide justification for current sampling. In addition, if information was obtained from the pilot on why it was not sustainable that information should have fed into the interview guide questions and analysis plan.

Study setting: country, income level, characteristics of patients (insurance status, educational level), % current compliance. What are the differences between community and academic center (at organizational, patient, inner/outer setting levels) that justified use of these two settings

Data collection:

Issues regarding using hypothetical intervention as stated above

How were the CFIR constructs that the guide was based on chosen?

RESULTS

Define CL as clinician and AD as administrator

Instead of dividing it into barriers/facilitators may be more helpful to follow if divided into current barriers and factors associated with implementation effectiveness

Table 1: I think dividing things as described above would make things more clear. A brief definition of what each domain and construct refer to would be helpful. The illustrative quotes are helpful but what would be more helpful is a brief description of the emergent theme around adaptability, complexity, etc. Just the word is not helpful when looking at the table.

Table 2 is a good layout but again under barriers there is often just the domain: construct listed not a description or theme that emerged as a barrier. Would be good to have that column remain consistent with the solutions column.

For those not familiar with “external pharmacies” a brief description of this would be helpful.

DISCUSSION:

First paragraph: consider reframing as stated above

Policies/procedures/standardization:

Figure 1: needs a legend, at this time difficult to follow. IS this what happens currently or what the intervention will entail? Or what the hypothetical intervention looks like? I thought the latter but the preceding sentence states this is the aspects of existing OAA, so confusing. Things labeled as moderators seem to be things that may make adherence more or less difficult. Perhaps using alternate terminology to “moderators” such as factors influencing adherence?

Acronyms should be defined

6. PLOS authors have the option to publish the peer review history of their article (what does this mean?). If published, this will include your full peer review and any attached files.

Reviewer #1: No

---

## [Author Response · Author response to Decision Letter 0]

24 Mar 2023

First, we highlight the following major points addressed by the reviewers:

• The reviewer recommended that the results are reorganized from being either barriers or facilitators to being either current barriers and facilitators or anticipated barriers and facilitators of adherence monitoring. They also recommended shifting wording in the introduction and discussion sections to reflect this change of scope of the manuscript. We greatly appreciate this feedback from the reviewers. However, we feel that organizing the barriers and facilitators into the categories of “current” and “anticipated” would result in the scope of this manuscript becoming too focused on the barriers and facilitators of a very specific OAA intervention. By keeping the manuscript broad in scope, the findings can be adapted to OAA interventions designed in other settings. Therefore, we chose to leave the results in the current “Barriers” and “Facilitators” format. However, we did agree with the reviewer that labeling each barrier and facilitator as “current” or “anticipated” in the current format is confusing and narrow in scope, and we removed these labels.

• The reviewer noted that this manuscript heavily referenced prior interventional work throughout the introduction, adding confusion to whether the interviews and identified barriers and facilitators were specifically for this intervention or for a theoretical intervention. We agree that the in depth description of our previous work added confusion regarding the scope of the identified barriers and facilitators in this manuscript. As a result, we have broadened the focus of the introduction section to encompass the work and recommendations of multiple OAA interventions. We have also removed the reference in the methods section suggesting that the structured OAA adherence intervention described in interviews was based on our prior work, as it was based on this collection of previous OAA interventions.

• The reviewer recommended including details on the differences between the community and academic centers at which individuals were interviewed. They also recommended interviewing individuals who had previous experience with an OAA intervention. We agree that this information is critical to applying our results to future OAA interventions. Therefore, we differentiated within the manuscript that the reason for choosing these two different sites was due to differences in resources addressing oral chemotherapy at each location. We clarified that those interviewed at the academic medical center had experience with a prior OAA adherence program while those at the community medical center did not. We also included the requested characteristics of patients treated at these hospitals where available.

Second, we highlight the following minor points addressed by the reviewers:

• The reviewers requested that definitions for each CFIR construct are added to the manuscript and Table 1. We greatly value this feedback from the reviewers. However, CFIR constructs and the process for identifying these constructs has been previously defined and published. We have provided the citation to this work within the manuscript and Table 1 for clarification.

• The reviewers had several requests in terms of reformatting the manuscript to match the journal’s requirements. We appreciate this feedback and have made the necessary adjustments, including adjusting the font, adding the tables to the main manuscript document, updating formatting, and adding a caption to Figure 1.

• The reviewers noted that the funding statement did not match the COI statement. We appreciate this feedback and have updated the funding statement appropriately, using the recommended language from the journal.

---

## [Decision Letter · Decision Letter 1]

24 Apr 2023

PONE-D-22-22462R1Barriers and facilitators associated with implementing interventions to support oral anticancer agent adherence in academic and community cancer center settingsPLOS ONE

Dear Dr. Muluneh,

Thank you for submitting your manuscript to PLOS ONE. After careful consideration, we feel that it has merit but does not fully meet PLOS ONE’s publication criteria as it currently stands. Therefore, we invite you to submit a revised version of the manuscript that addresses the points raised during the review process.

I commend on the authors' careful consideration of the last review comments and their appropriate revisions. We unfortunately was not able to engage the initial reviewers to review the revisions. The new reviewer is a very experienced researcher in cancer research in primary health care setttings. She is complimentary too of the revisions and have generously provided the authors a few more suggestions to improve the manuscript. I encourage the authors to provide further minor revisions and I will make a final decision accordingly.

We look forward to receiving your revised manuscript.

Kind regards,

Phyllis Lau, PhD

Academic Editor

PLOS ONE

Journal Requirements:

Additional Editor Comments :

I commend on the authors' careful consideration of the last review comments and their appropriate revisions. We unfortunately was not able to engage the initial reviewers to review the revisions. The new reviewer is a very experienced researcher in cancer research in primary health care setttings. She is complimentary too of the revisions and have generously provided the authors a few more suggestions to improve the manuscript. I encourage the authors to provide further minor revisions and I will make a final decision accordingly.

Reviewers' comments:

Reviewer's Responses to Questions

**Comments to the Author**

1. If the authors have adequately addressed your comments raised in a previous round of review and you feel that this manuscript is now acceptable for publication, you may indicate that here to bypass the “Comments to the Author” section, enter your conflict of interest statement in the “Confidential to Editor” section, and submit your "Accept" recommendation.

Reviewer #2: All comments have been addressed

2. Is the manuscript technically sound, and do the data support the conclusions?

Reviewer #2: Yes

3. Has the statistical analysis been performed appropriately and rigorously? 

Reviewer #2: N/A

4. Have the authors made all data underlying the findings in their manuscript fully available?

Reviewer #2: Yes

5. Is the manuscript presented in an intelligible fashion and written in standard English?

Reviewer #2: Yes

6. Review Comments to the Author

Reviewer #2: This is a nice update from the first iteration of the paper. A few minor points: make sure to explain the limitations of using people you know as the sample, this is important to address in the results - who were snowballed sampled as well? Can you include what the potential sampling bias might be in the Discussion?

Be careful with some of your language 'living below poverty' is this 'living in poverty' or 'below the poverty line'? Define 'PDC' in the table.

More importantly I like the introduction of the Table but can you make it more comprehensive to reflect the results? At this stage the quotes are examples of each construct and the table doesn't stand alone as a precis of the results and findings.

7. PLOS authors have the option to publish the peer review history of their article (what does this mean?). If published, this will include your full peer review and any attached files.

Reviewer #2: **Yes: **JENNIFER MCINTOSH

---

## [Author Response · Author response to Decision Letter 1]

9 May 2023

April 28, 2023

Dear Editorial Board:

It is with great enthusiasm that I submit this revised submission for the manuscript titled “Barriers and facilitators associated with implementing interventions to support oral anticancer agent (OAA) adherence in academic and community cancer center settings” to your esteemed journal. We would like to thank the reviewers for their critiques and their time identifying and articulating potential weaknesses of the manuscript. We have addressed these concerns and made relevant changes to improve the paper.

Here we address the comments raised by the reviewer:

(1) “Make sure to explain the limitations of using people you know as the sample, this is important to address in the results - who were snowballed sampled as well? Can you include what the potential sampling bias might be in the Discussion?”

Response: We have now added a few sentences to address this limitation in the discussion section. The reviewer is correct that there is inherent bias in conducting this work in the same institution and having some prior connections with interviewees. We tried to mitigate this by having all of the interviews conducted by a research staff who did not have prior knowledge of the individuals interviewed. 

(2) “Be careful with some of your language 'living below poverty' is this 'living in poverty' or 'below the poverty line'?”

Response: Thank you for this comment. It should read “living below the poverty line” which we have now corrected. 

(3) “Define 'PDC' in the table.”

Response: Thank you for this comment. We have spelled out PDC in the table. 

(4) “More importantly I like the introduction of the Table but can you make it more comprehensive to reflect the results? At this stage the quotes are examples of each construct and the table doesn't stand alone as a precis of the results and findings.”

Response: Thank you for this comment. We modified the reference to the tables as follows: “Barriers and, facilitators (along with illustrative quotes), and as well as potential proposed solutions by interviewees are outlined below and briefly summarized in Tables 1 and 2.”

Thank you again for the insightful feedback raised above. We hope we have appropriately addressed the reviewer’s comments. Thank you for the opportunity to resubmit our manuscript. 

Sincerely,

Benyam Muluneh, PharmD, BCOP, CPP

Assistant Professor, UNC Eshelman School of Pharmacy

---

## [Editor Report · Decision Letter 2]

22 May 2023

Barriers and facilitators associated with implementing interventions to support oral anticancer agent adherence in academic and community cancer center settings

PONE-D-22-22462R2

Dear Dr. Muluneh,

We’re pleased to inform you that your manuscript has been judged scientifically suitable for publication and will be formally accepted for publication once it meets all outstanding technical requirements.

Kind regards,

Phyllis Lau, PhD

Academic Editor

PLOS ONE

Additional Editor Comments (optional):

Thank you for persevering and providing considered responses to the last reviewers' comments. My congratulations!

---

## [Editor Report · Acceptance letter]

14 Jul 2023

PONE-D-22-22462R2 

Barriers and facilitators associated with implementing interventions to support oral anticancer agent adherence in academic and community cancer center settings 

Dear Dr. Muluneh:

I'm pleased to inform you that your manuscript has been deemed suitable for publication in PLOS ONE. Congratulations! Your manuscript is now with our production department. 

Kind regards, 

on behalf of

Dr. Phyllis Lau 

Academic Editor

PLOS ONE